# Hybrid Value-Aware Transformer Architecture for Joint Learning from Longitudinal and Non-Longitudinal Clinical Data

**DOI:** 10.3390/jpm13071070

**Published:** 2023-06-29

**Authors:** Yijun Shao, Yan Cheng, Stuart J. Nelson, Peter Kokkinos, Edward Y. Zamrini, Ali Ahmed, Qing Zeng-Treitler

**Affiliations:** 1Department of Clinical Research and Leadership, School of Medicine and Health Sciences, George Washington University, Washington, DC 20037, USA; yan_cheng@gwu.edu (Y.C.); stunelson@gwu.edu (S.J.N.); peter.kokkinos@va.gov (P.K.); zamrinie@gwu.edu (E.Y.Z.); ali.ahmed@va.gov (A.A.); zengq@gwu.edu (Q.Z.-T.); 2Washington DC VA Medical Center, Washington, DC 20422, USA; 3Department of Kinesiology and Health, School of Arts and Sciences, Rutgers University, New Brunswick, NJ 08901, USA; 4Department of Neurology, School of Medicine, University of Utah, Salt Lake City, UT 84112, USA; 5Irvine Clinical Research, Irvine, CA 92614, USA; 6Department of Medicine, School of Medicine, Georgetown University, Washington, DC 20057, USA

**Keywords:** deep learning, longitudinal data, Alzheimer’s disease and related dementias, disease risk prediction

## Abstract

Transformer is the latest deep neural network (DNN) architecture for sequence data learning, which has revolutionized the field of natural language processing. This success has motivated researchers to explore its application in the healthcare domain. Despite the similarities between longitudinal clinical data and natural language data, clinical data presents unique complexities that make adapting Transformer to this domain challenging. To address this issue, we have designed a new Transformer-based DNN architecture, referred to as Hybrid Value-Aware Transformer (HVAT), which can jointly learn from longitudinal and non-longitudinal clinical data. HVAT is unique in the ability to learn from the numerical values associated with clinical codes/concepts such as labs, and in the use of a flexible longitudinal data representation called clinical tokens. We have also trained a prototype HVAT model on a case-control dataset, achieving high performance in predicting Alzheimer’s disease and related dementias as the patient outcome. The results demonstrate the potential of HVAT for broader clinical data-learning tasks.

## 1. Introduction

Transformer is a deep neural network (DNN) architecture for learning from sequence data with natural language data as a primary example. Since its invention in 2017 [1], Transformer has become the state-of-the-art approach in many natural language processing (NLP) tasks due to its excellent performance [2,3,4,5,6,7]. The Transformer architecture is based on an attention mechanism, which allows the model to focus on different parts of the input sequence, thus capturing long-range dependencies and improving the quality of predictions.

One of the most successful applications of Transformer in NLP is the Bidirectional Encoder Representations from Transformers (BERT) model [2]. BERT is a pre-trained language model that has achieved state-of-the-art performance on a wide range of NLP tasks, including text classification, question-answering, and named entity recognition. Another popular Transformer-based language model is the Generative Pre-trained Transformer (GPT) [3,4,5], which has been used for various language generation tasks, such as machine translation and text summarization. The recently emerged chatbot known as ChatGPT, which has gone viral on the internet since its launch [8], was developed based on GPT to generate human-like responses to user input [9].

The success of Transformer-based models in NLP has inspired researchers to explore their application in domains outside NLP, such as healthcare. Longitudinal clinical data, which reside mostly in electronic health records (EHR), are also a type of sequence data and are, in many ways, similar to natural language data [10,11]. In natural language, a sentence, for example, can be viewed as a sequence of words in which the order of the words impacts its meaning. Thus, the positions of the words in a sentence are a critical component of the data. In clinical data, coded and/or textual data are recorded along with timestamps. Clinical concepts, which can be either defined using codes or extracted from the texts to represent diagnoses, medications, lab tests, vital signs, etc., play a similar role to the words in a sentence. The timestamps, which determine the temporal order of the clinical concepts, play a similar role to the word positions. Hence, it is a natural question whether the Transformer architecture can also be applied to clinical data and achieve excellent learning performance as well.

To answer this question, we must be aware that clinical data are much more complex than natural language data in several ways: (1) the time gaps between consecutive codes/concepts are irregular, while the position gaps between consecutive words are always one; (2) multiple codes/concepts may share the same time point, while no two words take the same position in a sentence; (3) a code/concept can have an associated numerical value (e.g., a lab test has a lab value), while no words have associated values; and (4) there are non-longitudinal data that are often needed for learning in addition to longitudinal data. These differences between clinical data and natural language data have made it a challenge to apply Transformer to clinical data for effective learning.

It is worth noting that there are Transformer-based clinical NLP models, which were obtained by fine-tuning the BERT model using clinical texts as specialized natural language data [12,13]. These models are designed to deal only with unstructured text data, while we focus more on structured clinical data, including structured raw EHR data and data processed from both structured and/or unstructured raw data and presented in a structured form.

In recent years, several Transformer-based DNN architectures for structured EHR data learning were developed, which partially addressed the differences between clinical data and natural language data. Li et al. proposed an architecture named BEHRT for pre-training using unlabeled data, which were further trained for disease prediction [14]. Their model used only diagnoses and patient ages as input data. They represented patient data as a sequence of diagnostic codes, together with sequence orders of the visits and patient ages at the visits. Patient age was used both as a risk predictor and a source of temporal order information for the diagnoses. Their embedding layer used multiple types of embedding, including concept embedding (for diagnoses), positional encoding (for visit sequence orders), age embedding, and segment embedding (for distinguishing adjacent visits). Rasmy et al. similarly designed Med-BERT but for learning from a much larger vocabulary of diagnostic codes [15]. In addition, they removed the use of ages; hence their data representation contained no temporal order information but only sequence order information. Pang et al. introduced CEHR-BERT as an improvement on both BEHRT and Med-BERT [16]. They expanded the vocabulary to include not only diagnoses but also medications, procedures, etc. and used both age embedding and time embedding to incorporate time information for each concept in the sequence. Kodialam et al. proposed SARD architecture, which was also inspired by BEHRT [17]. Their embedding is at the visit level, with each visit embedding being the sum of all embeddings of the concepts recorded during that visit. They used temporal encoding similar to the positional encoding in the original Transformer model to incorporate time information.

In this paper, we present a novel Transformer-based DNN architecture for joint learning from both longitudinal and nonlongitudinal clinical data. We referred to it as the Hybrid Value-Aware Transformer (HVAT). The word “hybrid” here refers to that the architecture is not a pure Transformer but a Transformer combined with another type of neural network. We also present a proof-of-concept experiment where a prototype HVAT model was developed using a dataset created in a prior study about Alzheimer’s Disease and Related Dementias (ADRD) to demonstrate the use and the capability of HVAT.

## 2. Materials and Methods

### 2.1. HVAT Architecture

Based on the original Transformer architecture [1], we design the HVAT architecture for learning from both longitudinal and non-longitudinal clinical data. The HVAT architecture has 2 branches at the input end to receive longitudinal and non-longitudinal data, respectively (Figure 1). The main branch, taking longitudinal data as input, is called a Value-Aware Transformer (VAT). The second branch, taking non-longitudinal data as input, is a feed-forward neural network (FFNN) with a residual connection [18]. The 2 branches join together at their last layers through a summation operation, followed by another FFNN with a residual connection, and lastly, followed by the output layer, which is a single-node layer with the sigmoid function as the nonlinear activation function. The sigmoid function is defined as $\sigma \left(x\right)=e^{x}/\left(1+e^{x}\right)$. The output is a single value p between 0 and 1. This type of output layer can be used for predicting binary outcomes coded by 0 and 1, respectively. The adverse outcome is usually coded by 1. The loss function is the binary cross-entropy function.

### 2.2. Input Data Representation

Non-longitudinal clinical data are represented in the usual tabular format, i.e., 1 vector per patient, and the vectors are all of the same dimension. Such a representation is commonly used in traditional statistical modeling (e.g., logistic regression) and traditional machine learning (e.g., support vector machine).

For longitudinal clinical data, the representation is inspired by how the natural language data is represented and used by the original Transformer model. In natural language, a sentence is represented as a sequence of word tokens, and each word token is simply a word paired with its position in the sentence. Therefore, the repeated words in a sentence are considered different tokens. For example, in the sentence “to go or not to go”, there are 4 words (i.e., “to”, “go”, “or”, “not”) but 6 tokens:

(1, “to”), (2, “go”), (3, “or”), (4, “not”), (5, “to”), (6, “go”).

In general, word tokens generated from a sentence are written in the form of (i,w), where i is the position of the word w in the sentence. Longitudinal clinical data are represented in a similar format through the following steps.

First, for each patient, we specify a time window for the patient history. We refer to the endpoint of the time window as the *index time/date*. The time window is divided into smaller equal-length intervals, and the intervals are indexed by natural numbers 1, 2, 3, …, from the latest to the earliest. These natural numbers are called *temporal indices*. The time window can vary from patient to patient, but the length of the small intervals is always the same for all patients. This step effectively normalizes the time information of the longitudinal data for all patients. How to choose the length of the intervals should depend on the specific tasks. For example, if the task is to predict outcomes for patients in an intensive care unit (ICU) stay, an appropriate choice for the length may be at the scale of an hour to a day. If the task is to predict a disease that takes years to develop, then a good choice for the length may be at the scale of a month to a year.

Next, for each patient, the longitudinal data from the time window specified for the patient is represented as a sequence of clinical tokens, whose length may vary by patient. A *clinical token* is a triple (t,C,v), where t is a temporal index, C is a clinical concept, and v is either a numerical value associated with C or the default value zero. A *clinical concept* is defined as a clinically meaningful feature or variable identified or extracted from the clinical data. Examples of a clinical concept include a diagnosis (which may be represented by 1 diagnostic code or a group of related diagnostic codes), the prescription and use of a medication, a lab test (which may be represented by 1 lab code or a group of lab codes), a hospitalization, etc. Some clinical concepts have associated numerical values (e.g., lab tests have lab values), while others do not (e.g., diagnoses do not have associated values). Even if some clinical concepts have associated values in records, they can still be used as concepts without values. If C is a clinical concept with associated values, then v in the triple (t,C,v) is the associated value of C at the temporal index t. If C is a clinical concept without values, then v takes the fixed value 0. If a concept C occurs multiple times over the same time interval with index t, only 1 token for this concept is generated for the temporal index t, and if the multiple occurrences of this concept have multiple associated values, then the associated value v in the clinical token (t,C,v) is the value aggregated from the multiple values. The aggregation method depends on the particular concept and application. Below we give an example of representing longitudinal clinical data as clinical tokens.

Example: Suppose the time window for a patient is the calendar year of 2012; hence the index date is 31 December 2012. Within the time window, the patient had 2 diagnoses, diabetes and hypertension, on 19 January 2012; 2 lab tests, calcium and glucose, on 15 April 2012, with values of 9.5 and 199, respectively, and 3 diagnoses, diabetes, hypertension, and a-fib, on 8 October 2012. We choose the length of the time interval to be 1 month (or 30.5 days). Then the sequence of clinical tokens for this patient is: 

(3, diabetes, 0), (3, hypertension, 0), (3, a-fib, 0), (9, calcium, 9.5), (9, glucose, 199), (12, diabetes, 0), (12, hypertension, 0).

Comparing the sequence of clinical tokens generated from the clinical data of a patient with the sequence of word tokens generated from a sentence in natural language, we see both similarities and differences (Table 1). Therefore, the representation of longitudinal clinical data is similar to but also more complicated than the representation of natural language data.

For each patient, a special token is added to the sequence of clinical tokens generated from the patient’s data. The special token takes the form (0,S,0), where S is an artificial concept defined to be different from all the clinical concepts. It plays the same role as the [CLS] token used in the BERT model [2]. The output corresponding to the special token by the VAT branch will be a summarization of all the clinical tokens in the sequence, which is made possible by the attention layers in the Transformer blocks. Then this output, which is always a vector of a preset dimension regardless of the sequence length, and the output by the FFNN branch are summed together at the summation layer where the 2 branches join.

### 2.3. Token Embedding

When a clinical token (t,C,v) is fed into the VAT branch, it is first transformed by the token embedding layer into a vector of dimension d. Token embedding is defined based on 3 more basic embeddings: temporal embedding, concept embedding, and value embedding (Figure 1). Their specific definitions are given below: 

(1) The temporal embedding, which maps each temporal index t to a vector TE(t) of dimension d, is defined by
$$TE\left(t\right)=[\sin\left(t\omega_{1}\right),\;\cos\left(t\omega_{1}\right),\;\sin\left(t\omega_{2}\right),\;\cos\left(t\omega_{2}\right),\;\cdots ,\;\sin\left(t\omega_{d/2}\right),\;\cos\left(t\omega_{d/2}\right)],\;t=0,\;1,\;2,\;\cdots$$
where $\omega_{k}={10,000}^{-2k/d}$ for k=1,⋯,d/2 (d is preset to be an even number). Note that the temporal embedding is not changed during the learning process.

(2) The *concept embedding*, which is learned from the data, maps each concept C to a vector CE(C) of dimension d.

(3) The *value embedding*, which is also learned from the data, maps each concept C to a second vector VE(C) of dimension d. The purpose of the value embedding is to provide a basis vector to incorporate the value associated with C into the token embedding.

The token embedding of the token (t,C,v) is defined by
E(t,C,v)=TE(t)+CE(C)+v×VE(C)

The token embeddings of all the tokens in a sequence are fed into the VAT branch simultaneously (Figure 1). In practice, v should be concept-wise normalized values rather than the raw values in the token embedding.

### 2.4. Dataset and Cohort

The data source was the U.S. Veterans Affairs (VA) Corporate Data Warehouse, a nationwide database for EHRs of U.S. Veterans. We used the dataset created in a prior study which was focused on the association between physical fitness and ADRD risks [19]. The study adopted the exercise treadmill test (ETT) as the indicator for physical fitness, which was measured in metabolic equivalents (METs) (1 MET = 3.5 mL of oxygen utilized per kilogram of body weight). A natural language processing (NLP) model was developed [20] and applied to all the medical notes in the database to extract MET values contained within those notes. For the initial cohort, we identified over 0.8 million patients who had at least 1 assessment of ETT with a value between 2.0 METs and 23.9 METs.

We employed a case-control design. For the case group, we sampled from the initial cohort 50,000 patients who received an ADRD diagnosis on or before 31 December 2019. For the control group, we randomly sampled 50,000 Veterans from the initial cohort who were alive on 31 December 2019 and were free of ADRD diagnoses by that date. The final cohort was defined to be the combination of the case group and the control group, having a total of 100,000 patients. We defined the endpoint to be the date of the first ADRD diagnosis for each case and to be 31 December 2019 for each control. Then we defined the index date to be 3 months prior to the endpoint for everyone. The time window was defined to be from 1 January 2000 up to the index date. The time window was divided into intervals of 1 year long for the temporal index assignment. To develop the HVAT model, we randomly split the cohort into 3 subsets: training (80%), validation (10%), and testing (10%), where each subset has an equal number of cases and controls.

### 2.5. Data Preparation

The outcomes were the case statuses, which were coded as: case = 1 and control = 0. The predictors from non-longitudinal data included age (at index date), sex, race, and ethnicity. The predictors from longitudinal data included MET, body-mass index (BMI), 15 diagnoses, 400 medications, and 400 note titles.

Age was used as a numerical variable. Sex was a binary variable coded as: female = 1 and male = 0. Race was a multi-category variable with 4 categories: Black, White, Other, and Unknown. Taking White as the reference category, we converted race into 3 dummy variables named Race_Black (vs. White), Race_Other (vs. White), and Race_Unknown (vs. White). Ethnicity was a multi-category variable with 3 categories: Hispanic, Non-Hispanic, and Unknown. Taking Non-Hispanic as the reference category, we converted ethnicity to 2 dummy variables named Ethnicity_Hispanic (vs. Non-Hispanic) and Ethnicity_Unknown (vs. Non-Hispanic). So, the non-longitudinal data were represented as vectors of 7 dimensions corresponding to the 7 variables.

Among the features from longitudinal data, only MET and BMI were used as clinical concepts with values; all other features were used as clinical concepts without values. When there were multiple BMI values over the same time interval, we used the mean function for aggregation. When there were multiple MET values over the same time interval, we used the maximum function for aggregation. The 15 diagnoses were defined based on manually selected ICD-9-CM and ICD-10-CM codes. The 400 medications and 400 note titles were selected out of the total of thousands of medications and note titles using a feature selection method described below.

### 2.6. Feature Selection

Since both medications and note titles were used as clinical concepts without values, the information they carry is in their presence-absence status. For simplicity, we only considered the presence-absence status over the entire time window for each patient.

For each feature (a medication or a note title), we first calculated the 2 × 2 contingency table as follows:

**# in Cases****# in Controls****Presence**ab**Absence**m−an−bwhere m and n are the total numbers of cases and of controls, respectively. Features with very distinct distributions in cases and controls are useful for prediction; for such a feature, the prevalence ratio (PR), defined as $\frac{a}{m}/\frac{b}{n}$, is expected to be distant from 1, or equivalently, the (natural) log of PR (LogPR) is expected to be distant from 0. 

Next, we calculated the adjusted estimates of LogPR and its standard error SE, respectively, using the Walters formula [21]:$$\mathrm{LogPR}=\log\left(\frac{a+0.5}{m+0.5}/\frac{b+0.5}{n+0.5}\right),\;\;\;\;\;\;\;\mathrm{SE}=\sqrt{\frac{1}{a+0.5}-\frac{1}{m+0.5}+\frac{1}{b+0.5}-\frac{1}{n+0.5}}$$

The equations were defined even when a or b was zero, thanks to the added small value of 0.5. Then the confidence interval of LogPR at (1−α)-level was
$$\left[\mathrm{LogPR}-z_{\alpha /2}\times \mathrm{SE},\;\;\mathrm{LogPR}+z_{\alpha /2}\times \mathrm{SE}\right]$$
where $z_{\alpha /2}$ is the z-critical value corresponding to α/2. 

We defined a feature ranking score using 1 of the 2 limits of the confidence interval: when LogPR≥0, the score was the lower limit, and when LogPR<0, the score was −1 times the upper limit. We found that this score could be expressed conveniently in 1 equation as
$$\mathrm{Score}=|\mathrm{LogPR}|-z_{\alpha /2}\times \mathrm{SE}$$
where | · | indicates the absolute value. Features with both a larger absolute value of LogPR and a smaller SE were ranked higher by this score. This score provides a balance between the 2 conditions, and the balance can be adjusted by changing α. A score less than 0 indicates that the LogPR is not significantly different from 0 at level 1−α.

In this study, we estimated the feature ranking scores using the training set only, which meant m=n=40,000. We chose α=0.05, so $z_{\alpha /2}=1.96$. Then we selected the top 400 medications and the top 400 note titles ranked by the scores for model development. All the selected features had a score greater than 0, which indicated that all of the selected features had a nonzero LogPR with statistical significance at a 95% confidence level.

### 2.7. Model Development

An HVAT model was developed using the final cohort to distinguish the cases from the controls. For the VAT branch, we used N=2 Transformer blocks. We used multi-head attention with 2 heads within the Transformer blocks. For the token embedding, we set the dimension d=32, which was also equal to the dimension of the output vector by the VAT branch. For all the dropout layers in the entire neural network, we set the dropout rate at 0.1. We used the rectified linear unit (ReLU) function [22] as the nonlinear activation function in the entire neural network, except that for the output node, it was the sigmoid function.

The training set was divided into mini-batches consisting of varying numbers of patients. Patients within the same mini-batch had similar numbers of tokens in their respective sequences, and the total number of tokens in the mini-batch was equal to or close to 10,000. Thus, the mini-batches with more patients all had shorter sequences and vice versa. Because of that, the mini-batch-wise loss function was defined as the sum of the loss functions over the patients within the mini-batch. The model was implemented in Python language using the PyTorch library. The weights in all the layers were initialized as small random numbers using K. He’s method [23]. The weights were updated during training using mini-batch stochastic gradient descent with Nesterov momentum [24]. We set different learning rates for different parts of the HVAT model: $5\times 10^{-4}$ for the FFNN branch, $5\times 10^{-5}$ for the VAT branch, and $1\times 10^{-4}$ for the remaining part of the model (i.e., the other FFNN together with the output layer in Figure 1).

The primary metric of model performance was the area under the receiver operating characteristics curve (AUC). The training process was stopped when the performance on the validation set plateaued, which was defined as no improvement in AUC over 10 consecutive epochs. The final model was set to be the 1 with the highest AUC before the plateau. Its performance on the testing set was reported as the model performance. The threshold on the output scores that maximized the accuracy on the training set was chosen to calculate the sensitivity, specificity, and accuracy on the testing set.

### 2.8. Comparative Study

In addition to the HVAT model, we also trained 4 other models for comparison. The first model was the HVAT without the FFNN branch, which was named “Without FFNN”. This model only used longitudinal data as input. The second model was the HVAT without the VAT branch, which was named “Without VAT”. This model only used non-longitudinal data as input. The third and the fourth models both used the full data but with longitudinal data converted to a non-longitudinal format so that the temporal order information in the longitudinal data was lost. The third model was a linear support vector machine (SVM) model, which was named “SVM”. The fourth model was the deep FFNN model obtained by removing the VAT branch from the HVAT model and increasing the number of nodes in the input layer to the total number of predictors so that it could take all the predictors’ (aggregated) values as input. We named the fourth model “FFNN-NL”. The SVM model could only capture the linear relationships between the predictors and the outcome, while the FFNN-NL model was able to capture non-linear relationships.

## 3. Results

The performance in AUC on the testing set for the five models from the comparative study is shown in Figure 2.

The performance of the model with only the longitudinal data as input (Without FFNN) was very close to that of the HVAT model, while the model with only the non-longitudinal data (Without VAT) was significantly lower. This suggests that the performance of the HVAT model was mainly attributed to the longitudinal data, which in turn shows that the HVAT model had learned relevant patterns from longitudinal data.

The SVM and FFNN-NL performed lower than HVAT by over 5 percentage points, suggesting that HVAT had learned important temporal information from the longitudinal data. That FFNN-NL was only slightly better than SVM showed that there was not a prominent nonlinearity between the (selected) predictors and outcome. Both SVM and FFNN2 performed relatively well (around 0.9), suggesting that the feature selection was helpful.

To report the performance of the HVAT model in other metrics, we first identified the threshold (which was 0.4443) on the output scores that maximized the accuracy on the training set. Then we calculated the corresponding sensitivities, specificities, and accuracies on the three subsets using the threshold. The results are listed in Table 2.

## 4. Discussion and Conclusions

In this study, we designed HVAT, a DNN architecture for joint learning from longitudinal and non-longitudinal clinical data. The HVAT was based on the original Transformer architecture designed for learning from natural language data, and our design leveraged the similarities while also addressing the differences between clinical data and natural language data. We also conducted a proof-of-concept experiment in which a prototype HVAT model was developed to classify the patients in an ADRD cohort. The model achieved an excellent performance. The comparative study showed that the HVAT model had learned relevant temporal information from the longitudinal data. It also showed that the HVAT model outperformed the conventional methods represented by the SVM model and the FFNN-NL model.

Our design is different from the other existing architectures aforementioned (in Introduction) in several ways. First, our architecture has a hybrid structure allowing for joint learning from both longitudinal and non-longitudinal data. The idea of using a hybrid structure for learning from data containing different modalities (e.g., longitudinal data and non-longitudinal data are two modalities) is not new since it has appeared in many previous studies [25,26,27,28]. However, when limited to the case of using Transformers for clinical data learning, the hybrid structure proposed in this study is new to the best of our knowledge. Second, our architecture can learn from the clinical concepts/codes with numerical values from the longitudinal data. Third, we used a simpler but more natural and more flexible longitudinal data representation and embedding method.

Since longitudinal clinical data are increasingly available as the EHR systems are widely adopted, and longitudinal data often contain important temporal information for outcome prediction, the ability to use longitudinal data is an advantage of the Transformer-based models over traditional machine learning models such as SVM as the latter only use data in a non-longitudinal format. However, the use of non-longitudinal data is still inevitable as there are important static data such as certain demographic data and genetic data. Here we need to stress that non-longitudinal data are not limited to static data. For example, age is not static, but since knowing the age at one time point makes it known at all time points, it is not necessary to use age as longitudinal data. We can just use age as non-longitudinal data in terms of, for example, age at index date, as in our experiment. Sometimes, one can also choose to convert part of longitudinal data as non-longitudinal data through an aggregation over time and use those for prediction. Therefore, we considered it necessary to jointly learn from both formats of data in clinical data learning.

The incorporation of the numerical values associated with clinical codes/concepts into the model is a key feature in our design because we realized that numerical values were common in clinical data and also crucial for outcome prediction. Examples of such values include lab values associated with lab tests, vital sign values associated with vital signs, drug dosages associated with prescribed drugs within a prescription, the length of stay associated with hospitalization, etc. One can be even more creative to come up with a non-obvious example: the total number of outpatient visits within a time interval corresponding to each temporal index. The clinical concept for such values may be called “visit frequency”.

In our design, the values are incorporated into the token embedding as multipliers to the value embedding vectors, which is essentially equivalent to treating the clinical concepts with values as continuous variables. We admit that there is at least one other way to utilize the values: dividing the range of values into multiple “bins” and then treating the concept with values from different bins as multiple different “concepts”. That way is similar to the common approach used in statistical modeling, which treats a continuous variable as a multi-categorical variable by discretizing the continuous values. We did not adopt the latter approach because we realized that it has certain disadvantages compared to our approach. First, the continuity of the values is lost due to the discretization. Bins that are close to each other and bins that are far from each other are treated in the same way. Second, it is often a difficult problem to discretize the continuous values. When the bins are too wide, there is a loss of information because different values in the same bins are treated as the same. When the bins are too narrow, there is a loss of power because some bins may contain too few examples. Third, there is usually no standard way to discretize all continuous variables, and many other considerations, including both clinical and mathematical, must be taken to do that on each. This would be time-consuming when there are many continuous variables to be used. 

Another key component of our design of HVAT is the use of temporal indices for representing the occurrence time of the clinical concepts. The temporal indices essentially give an order for the time intervals within the time window of the patient history. There are two possible ways to order the time intervals: forward and backward. We chose the backward order because the time windows of the patients are supposed to be aligned along the end of the time window rather than the start, which is also why the end of the time window is defined as the index time/date. This way, it makes the model easier to identify the relevant temporal patterns in the longitudinal data for predicting the outcomes.

The flexibility in customizing the length of the time intervals makes our architecture advantageous over the others (e.g., BEHRT). Actually, all the other architectures organize the longitudinal data at the visit level, which is equivalent to setting the time intervals to be one day long for HVAT (assuming at most one visit per day in most cases). However, for some tasks, such as the experiment we did in this study, where we used up to 20 years of history, lengthy sequences would be generated if the time intervals are set to be too short such as one day long. For example, assume an average patient has 10 visits per year and 10 concepts per visit, then over 20 years, the patient would have 2000 tokens in the sequence. If we set the time intervals to be one year long, then the patient would only have 200 tokens in the sequence. Considering that the computational time of Transformer is O(N2), where N is the sequence length, our architecture would consume only 1% of the computational time of the others. Actually, it was exactly by setting the time intervals to be one year long that we were able to complete the model training and inference in a reasonable time on an ordinary desktop computer while an excellent performance was still achieved. One problem caused by this choice of length was that the temporal order information of the clinical concepts occurring within the same interval would be lost. However, we believe that such a loss was negligible for a chronic condition such as ADRD. 

The flexibility on the length of time intervals also allows HVAT architecture to be applied to some situations where very short time intervals should be used. For example, in an ICU setting, the status of the patients may change from hour to hour, and to predict outcomes, it may be best to set the time intervals to be one hour (or a few hours) long. It would be difficult for architectures designed for visit-level data structures to be applied to such situations.

Most of the other Transformer-based models were designed to work like the NLP model called BERT [2], whose main strength is its ability to pre-train the model without human labeling. The HVAT architecture also allows such pre-training, although that was not demonstrated in our study. We believe that it is possible to design pre-training strategies similar to the “masked language model” and the “next sentence prediction” to pre-train an HVAT model on clinical data.

There are a few limitations in our study. First, we did not use certain important types of clinical concepts with values, such as lab tests in the prototype model from the experiment. The lab values contain valuable information for outcome prediction. However, we find that the lab values are usually messier than other types of data, such as diagnoses, and to use hundreds of lab tests as clinical concepts with values, it would take much effort and time to clean the lab data before they can be effectively used. Second, the prototype model is small in terms of the number of Transformer blocks, the dimension size of the embedding vectors, the number of attention heads, etc., compared to other Transformer-based models. This is mainly due to the limited computing resources available to us. However, the above design choices were sufficient for the purpose of prototyping.

For future work, we plan to develop an explaining method for the HVAT models, which can reveal the relevant temporal patterns learned by the model and apply it to risk factor analysis for ADRD and other adverse outcomes. We are also contemplating using HVAT to train a GPT-like model as a foundational model which can generate simulated longitudinal clinical data.

In conclusion, HVAT is a successful adaption of the Transformer architecture to clinical data, as it has addressed the complexities of the clinical data over the natural language data. The design and the results together have demonstrated the potential advantage of using HVAT in broader clinical data-learning tasks.

## Figures and Tables

**Figure 1 jpm-13-01070-f001:**
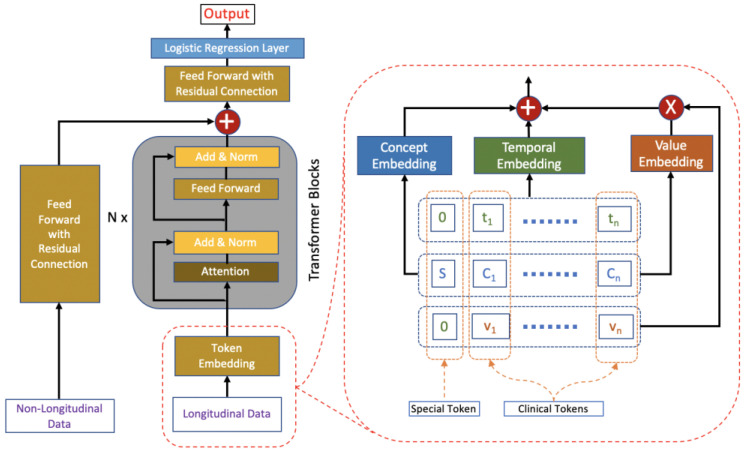
The HVAT architecture. The arrows indicate data flow directions. The left side (i.e., left to the words “Transformer Blocks”) is an overview of the HVAT architecture. The right side is a zoomed-in view of the part on the left side (enclosed by red dashed lines), which illustrates how the token embedding layer processes longitudinal data through three embeddings: concept embedding, temporal encoding, and value embedding.

**Figure 2 jpm-13-01070-f002:**
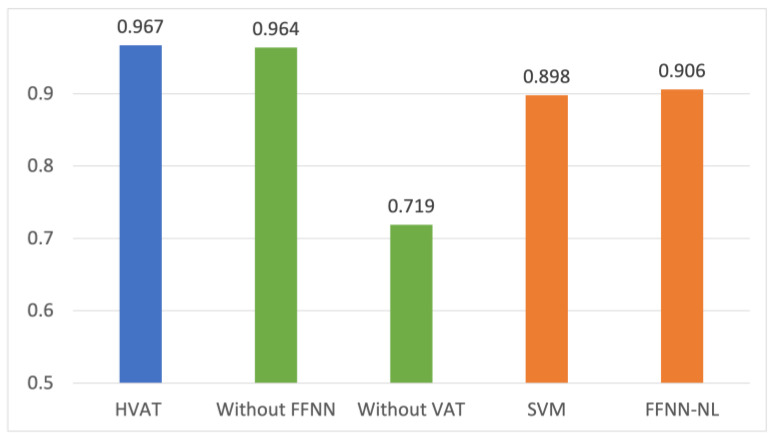
Comparison of the performance in AUC from the comparative study.

**Table 1 jpm-13-01070-t001:** A summarization of the similarities and differences between the representations of longitudinal clinical data and natural language data.

Similarities		Differences
The temporal index t is analogous to the word position i and the clinical concept C is analogous to the word w. The same clinical concept (resp. word) can occur at multiple but different temporal indices (resp. positions). Every combination of temporal index (resp. position) and clinical concept (resp. word) can occur at most once in the sequence.		Clinical concepts can have associated numerical values while words do not; For a sequence of word tokens, the positions are consecutive integers, with each integer occurring only once, while for a sequence of clinical tokens, the temporal indices from the sequence of clinical tokens are not necessarily consecutive integers: some integers can occur multiple times as temporal indices (though for different clinical concepts), and some integers may not occur at all.

**Table 2 jpm-13-01070-t002:** The performance of the full HVAT model on the testing set.

Sensitivity	Specificity	Accuracy
0.891	0.925	0.908

## Data Availability

Population-level aggregated data are available from the authors on reasonable requests.

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
