# Peer review of "Hybrid Value-Aware Transformer Architecture for Joint Learning from Longitudinal and Non-Longitudinal Clinical Data"

_jpm, 2023, doi:10.3390/jpm13071070_

Round 1
Reviewer 1 Report
Dear authors,
The hybrid model decribed by you in the manuscript is interesting and useful. However, there are some issues to be addressed.
1. The manuscript should be organized an appropriate way that creates interest to reader.
2. Introduction may be improved by discussing about topic elaborately.
3. Authors have quoted that it is a hybird model but they have not discussed comparision of newly developed hybrid model and and conventional methods.
4. There are any records of testing of the developed hybrid model.
Reviewer 2 Report
The authors present a novel Transformer-based deep neural network architecture called Hybrid Value-Aware Transformer (HVAT) that addresses the challenges of adapting the Transformer model to the complexities of clinical data. HVAT is capable of learning from both longitudinal and non-longitudinal clinical data, incorporating numerical values associated with clinical codes/concepts and utilizing a flexible longitudinal data representation called clinical tokens. The authors trained a prototype HVAT model on a case-control dataset and achieved high performance in predicting Alzheimer's disease and related dementias as patient outcomes, indicating the potential of HVAT for various clinical data learning tasks.
